# Using Potential Years of Life Lost (PYLL) to Compare Premature Mortality between Romanian Counties to Confirmed COVID-19 Cases in 2020 and 2021

**DOI:** 10.3390/healthcare12121189

**Published:** 2024-06-13

**Authors:** Diana Maria Rahotă, Dorel Petru Țîrț, Lucia Georgeta Daina, Cristian Marius Daina, Codrin Dan Nicolae Ilea

**Affiliations:** 1Faculty of Medicine and Pharmacy, University of Oradea, 1 December Sq., 410081 Oradea, Romania; 2Psycho-Neurosciences and Recovery Department, Faculty of Medicine and Pharmacy, University of Oradea, 1 December Sq., 410081 Oradea, Romania; 3Department of Surgical Disciplines, Faculty of Medicine and Pharmacy, University of Oradea, 1 December Sq., 410081 Oradea, Romania; 4Statistics Department, Bihor County Emergency Clinical Hospital, 67 Gheorghe Doja Street, 410169 Oradea, Romania

**Keywords:** potential years of life lost, PYLL, COVID-19, SARS-CoV-2, counties of Romania

## Abstract

This article examines the impact of the COVID-19 pandemic on potential years of life lost (PYLL) in Romania’s counties in 2020 and 2021. PYLL highlights the burden of premature deaths in a community and is a useful tool for prioritizing community health issues. The study compares the PYLL variation between different counties, identifying disparities in premature mortality rates and highlighting areas that require specific public health interventions. The results indicate that COVID-19 has had a significant impact on potential years of life lost across the country. For the year 2020, the total number of deaths from confirmed COVID-19 cases was 19,455, of which 14,152 premature deaths caused 193,489 PYLL, with a crude rate of 1053.28 PYLL per 100,000 inhabitants. In 2021, there were 39,966 deaths from confirmed COVID-19 cases, with 28,777 premature deaths, 386,061 PYLL, and a crude rate of 2116.63 PYLL per 100,000 population. This study reveals significant variations only in some counties, based on BYLL rates, and in the two years analyzed. The proportion of premature deaths (<80 years) varied by county and gender. PYLL’s analysis by gender shows that men experienced a higher number of premature deaths than women in most counties, and this trend persisted in both years. The results are presented in the form of thematic maps, highlighting standardized PYLL rates for both genders in each county, facilitating a visual understanding of regional disparities. The identified variations can serve as a basis for developing and implementing more effective public health policies, based on the specifics of each county.

## 1. Introduction

Potential Years of Life Lost (PYLL) rates are a useful tool for comparing the burden of specific causes of death between groups and highlighting the burden of premature deaths, which often have the greatest impact on a community [1]. PYLL is calculated by subtracting the age at death from that person’s standard life expectancy. This value helps highlight the years of life that are lost when a person dies prematurely, at the individual or community level, providing valuable information for public health interventions and policies.

Comparing potential years of life lost (PYLL) between counties in Romania is crucial for several reasons. First, it identifies disparities in premature mortality rates, which can highlight areas in need of targeted public health interventions and resource allocation [1]. By comparing PYLL between counties, it becomes possible to discern which regions are facing a higher burden of premature deaths and investigate the causes behind these discrepancies. This is particularly important in a country like Romania, where there are significant regional variations in terms of economic development and access to health services [2]. In addition, comparing PYLL can provide insights into the effectiveness of public health policies and interventions in different counties, helping to identify best practices and areas for improvement. In addition, comparing PYLL between counties in Romania can help in understanding the impact of certain health issues and diseases on different regions. For example, studies have shown variations in the prevalence of mosquito-borne pathogens and zoonotic diseases in different regions of Romania [3,4,5]. By comparing PYLL, it becomes possible to assess the impact of these diseases on premature mortality and prioritize public health measures accordingly. In addition, comparing PYLL between counties can shed light on the social, economic and environmental factors that determine health. Inequalities in access to health services, hygiene and preventive behavior have been documented in different regions of Romania [6,7,8,9]. Comparing PYLL can help understand how these disparities contribute to differences in preterm mortality rates and guide efforts to address these underlying determinants.

The COVID-19 pandemic has had a significant impact on Romania, with over 3,305,048 confirmed cases and 67,341 deaths reported by December 2022 [10]. The pandemic has led to various studies focusing on different aspects of the situation in Romania. For example, research has been conducted to understand attitudes towards COVID-19 vaccination among cancer patients [11], as well as the reasons for refusing vaccination against COVID-19 in the Romanian population [12]. In addition, the impact of the pandemic on different sectors, such as healthcare and education, has been explored. Studies have investigated the motivation of healthcare professionals and their work interests in the context of the COVID-19 pandemic [10], as well as teachers’ resilience and mindset regarding the use and efficiency of e-learning platforms during the pandemic period [13]. In addition, the economic and social implications of the pandemic have been a topic of interest. The impact of the pandemic on Romania’s economic growth was examined [14], together with an assessment of the SARS-CoV-2 infection rates among dental practitioners and the economic impact on dental practices [15]. In addition, the pandemic has led to a significant number of prematurely lost life years (PYLL) in Romania, with a study reporting 4,210,654 PYLL in 17 countries, including Romania, in August 2020 [16]. In addition, the spread of the COVID-19 virus in Romania was analyzed, with studies focusing on the early spread of the virus, cases imported from other countries and human-to-human transmission networks [17]. The impact of the pandemic on different regions in Romania has also been investigated, such as the introduction and characteristics of SARS-CoV-2 in the North-East region during the first COVID-19 outbreak [18]. Overall, the COVID-19 pandemic has had multiple effects on Romania, ranging from changing attitudes towards healthcare and vaccination to economic and social implications. The research conducted provides valuable insights into the different dimensions of the impact of the COVID-19 pandemic on the country.

The literature mainly contains information on the entire country and no articles presenting comparative aspects of the variation in PYLL between the territorial administrative units of Romania (counties and Bucharest Municipality) during the different waves of the COVID-19 pandemic have been identified.

In Romania, according to official data [19,20], 636,201 cases of people infected with the SARS-CoV-2 virus were confirmed in 2020, and cumulatively in 2020 and 2021, 1,810,342 cases of infection with the SARS-CoV-2 virus were registered. As of January 1, 2022, 58,779 people diagnosed with SARS-CoV-2 infection have died [20]. In Romania, in relation to the epidemiological situation, in order to prevent the spread of COVID-19 and manage its consequences [21], the authorities ordered the state of emergency (from 16 March 2020) and subsequently the state of alert (which, after several extensions, ended on 8 March 2022). The pandemic has had profound consequences on social life (through restrictions and containment measures) and on the health system (application of measures to prevent the spread of the disease, diagnosis and treatment of suspected and confirmed persons). During 2020 and 2021, Romania was affected by pandemic waves and registered, in certain periods, values for the number of daily confirmed cases per 1 million inhabitants higher than those in the European Union [22].

Even though there is controversy about the limitations of PYLL use [23,24], such as recording deaths with a particular cause, setting the norm for which the PYLL is zero, interpreting results, focusing on deaths, and ignoring quality of life, several ways of calculating BYL are described in the literature. The first modality is based on ‘life table norms for life years lost, which is the number of years lost by an ‘average’ person in a given population’ [25]. The second modality is to calculate the difference between the potential life limit and the age at death of the individual [26]. The sum of the years of life lost by each individual within a community is the total PYLL in that community (city, county, country). There is also the possibility of adjusting the PYLL by excluding it from the calculation cases with certain associated comorbidities [25,27]. For the comparability of data, ‘transparency in documenting the exact method used to calculate YLL’ [24] must be ensured, despite all existing and identifiable limitations. 

According to the OECD and the European Commission, the life expectancy at birth in Romania in 2019 was 75.6 years, and in 2020 it decreased to 74.2 years [28]. According to WHO/EURO, the life expectancy at birth in 2018 in Romania was 75.4 years (both genders), 79.1 years for women and 71.8 years for men [29]. Based on these data, the age limit for calculating PYLL for confirmed COVID-19 cases could be set at 75 years. In the literature on PYLL for COVID-19 deaths, it is recommended that the limit for premature death is set at 80 years [16,30]. For calculating the PYLL for comparability with international studies, in this study, we used the age limit of 80 years for premature deaths. 

The aim of the study is to investigate the variation in Potential Years of Life Lost (PYLL) caused by COVID-19 in Romanian counties in 2020 and 2021. This study aims to provide a comprehensive understanding of the impact of COVID-19 deaths on potential years of life lost in each county of the country. For each year studied, we will calculate the PYLL indicators: proportion of premature deaths out of total deaths in confirmed COVID-19 cases, number for deaths for each gender and the total for both genders by county, the crude rate of PYLL for each gender and the total for both genders by county, PYLL/death for each gender and the total for both genders by county, and the standardized rate of PYLL for both genders. The research was conducted using the model of a retrospectively descriptive epidemiological study.

## 2. Materials and Methods

### 2.1. Study Design

We conducted a retrospective descriptive observational epidemiological study of the deaths from confirmed COVID-19 cases in Romania and registered it in the Alerte.MS platform (national database). The analyzed deaths were registered between March 2020 and 21 April 2022. The data were used by researchers in an anonymized form.

### 2.2. Data Sources

Anonymized data on COVID-19 deaths at patient level were extracted from the Alerte.MS platform, which allows patient data to be exported from the Corona Forms platform. According to the legislation in force, the Corona Forms software application (version 4.3) is used by the Romanian Ministry of Health in order to “collect and correlate data provided by public authorities and institutions involved in monitoring SARS-CoV-2 virus infection, as well as to record confirmed persons and deaths produced by SARS-CoV-2 virus infection” [31]. For each death registered in the platform, the following fields were exported to an Excel database: age, gender, county of residence, county of infection, year of confirmation, date and time of confirmation, date and time of death, and healthcare-associated infection. The database of deaths registered in 2020, 2021 and 2022 (until 21 April 2022, according to Figure 1) included 65,406 cases, from which cases not mentioned in the county of infection (29 cases) and deaths registered in confirmed cases in 2022 (5956 cases) were excluded. Finally, confirmed cases in 2020 and 2021 (59,421 cases of death) were analyzed.

The numerical data on the resident population on 1 July 2020 and 1 July 2021 (for the 41 counties and Bucharest, Romania) in total and by 5-year age groups were obtained from the platform of the National Institute of Statistics of Romania [32].

The carrying out of this study was approved by the management of the Bihor Public Health Directorate, according to the document registered with the institution with no. 3937/19.03.2024.

### 2.3. Calculation of Indicators

Microsoft Excel was used to calculate the PYLL indicators. Premature deaths were considered those before the age of 80. From the death database, each death was assigned to the age group of 5 years corresponding to the age at death. For each death, the PYLL number was obtained by subtracting the age limit set (80 years) from the age limit at death.

Subsequently, for each county of Romania and Bucharest for each studied year (2020 and 2021), the total and by gender, the crude rate of PYLL and PYLL/death from confirmed COVID-19 cases were calculated.

The crude rate of PYLL was calculated according to the following formula:$$\mathrm{Crude}\;\mathrm{rate}\;\mathrm{of}\;\mathrm{PYLL}=\frac{\mathrm{PYLL}}{\mathrm{Number}\;\mathrm{of}\;\mathrm{population}<80\;\mathrm{years}}\times 100.000$$

Obtaining PYLL/death was achieved by dividing PYLL by the number of deaths that determined the respective potential years of life lost.

For calculating the standardized rate of PYLL for both genders and each county, we used direct standardization based on five-year age groups (0–4, 5–9, ...., 75–79) and the European population as the standard population [33]. 

Data processing and analysis were carried out with Microsoft Excel, starting from the anonymized database. The thematic map representation of standardized rates was performed with QGIS Desktop version 3.28.5 [34]. 

Statistical analysis was performed using the R program, version 4.3.3 [35]. The chi-squared test was used to compare categorial data, and the *t*-test was used to compare continuous variables.

## 3. Results

The number of deaths and PYLL indicators by Romanian counties in both genders are presented in Table 1 (for 2020) and Table 2 (for 2021).

The comparison of PYLL/death in 2020 versus 2021 shows an increase for the following counties: Bacau, Bistrita-Nasaud, Braila, Caras-Severin, Cluj, Constanta, Covasna, Galati, Gorj, Harghita, Ialomiţa, Maramureş, Satu Mare and Timiş. Statistical significance is recorded only for Covasna County (from 10.28 in 2020 to 14.03 in 2021, test *t*, *p* < 0.0001) and Galați County (from 12.51 in 2020 to 14.01 in 2021, test *t*, *p* < 0.05). For most counties, as well as nationally, a downward trend in PYLL/death was observed in 2021 compared to 2020. Statistical significance was found in the counties of Arges, Buzău, Teleorman (test *t*, *p* < 0.05) and at national level (decrease from 13.67 in 2020 to 13.42 in 2021, test *t*, *p* < 0.05). 

The proportion of premature deaths in total deaths from confirmed COVID-19 cases, for both genders, for each of the analyzed years, namely 2020 and 2021, is shown in Figure 2.

The proportion of premature deaths (<80 years) from confirmed COVID-19 cases per county/Bucharest municipality for both genders in the two analyzed years is presented in Figure 3 (2020) and Figure 4 (2021).

The PYLL indicators by gender (number of premature deaths, total PYLL count, crudePYLL rate per 100,000 population and PYLL/death) are presented in Table 3 (year 2020) and Table 4 (year 2021). 

The comparison of PYLL/death for female versus male shows that there is, at the national level, a statistically significant difference of over one year/death for men, both in 2020 (Table 3) and 2021 (Table 4). 

The standardised rates of PYLL for both genders by county are presented in map form for each of the years analyzed, namely 2020 (Figure 5) and 2021 (Figure 6).

The contribution of large age groups to premature mortality as a proportion of total premature mortality is shown in Figure 7 (number of premature deaths) and Figure 8 (PYLL).

In both years analyzed, the highest proportion of premature deaths in the 65–79 age group (66% in 2020 and 67% in 2021) is noted, while from the PYLL perspective, the 40–64 age group is the most affected (55% in 2020 and 54% in 2021).

## 4. Discussion

COVID-19 has resulted in a potentially significant years of life lost (PYLL) in various countries. In 2020, the PYLL per death rate ranged from 2.7 years in Australia to 19.3 years in Ukraine [16]. In Portugal, COVID-19 accounted for 4.2% of all years of life lost, with an exceedance of 35,510 Years of Life Lost (YLL or PYLL); of this, 72% was caused by COVID-19 and 28% by non-COVID-19 causes [36]. According to a study from Czechia (2022), complete vaccination reduced the number of potential life years lost to death by approximately 3.5 times compared to the number of expected deaths between October and December 2021 [37]. A study conducted in 34 countries found that COVID-19 caused 9 to 21 years of life lost per deceased patient, with East Asia and Oceania having fewer YLL per capita than North America and Europe [38]. In Turkey, men had 205,177 potential life years lost and women had 125,330 years lost due to COVID-19 in the first year of the pandemic [39]. In Hungary, the years of life lost to COVID-19 were shorter than expected when comorbidities were considered. The usual calculation shows 10.5 years of life lost for each death, which drops to 9.2 years per death after adjusting for comorbidities. But further research is needed to interpret this finding [25]. Another study (2022) conducted in 20 countries found that the burden of premature mortality from COVID-19 extends beyond older age groups, with some countries experiencing a high burden in younger groups [40]. These variations in PYLL highlight the need to protect vulnerable populations, especially older people, through careful public health measures [38]. Romania has experienced (from the beginning of the COVID-19 pandemic until 1 November 2020) a significant gender difference in the impact of the disease, with an average age higher than the age of death for women who died of COVID-19, leading to a decrease in the female mortality rate per death (18.84 for males and 23.12 for females). In addition, there were more male than female deaths in absolute numbers [41].

The data provided present information on the deaths and PYLL indicators (Potential years of life lost) related to confirmed COVID-19 cases in Romania for the years 2020 and 2021. The tables provide a breakdown by county, for the number of deaths, the number of premature deaths, the proportion of premature deaths in confirmed COVID-19 cases, PYLL, the crude rate of PYLL (total and by gender), the standardized rate of PYLL (both genders) and PYLL per death (total and by gender). 

In 2020, a total of 19,455 deaths from confirmed COVID-19 cases were recorded in Romania, with 14,152 premature deaths, resulting in a PYLL of 193,489 and a PYLL per death ratio of 13.67 (Table 1). Our study determined that the range of PYLL/death variation in Romanian counties in 2020 spans from 10.28 years (Covasna County) to 16.08 years (Ilfov County). A study carried out in 17 countries for the period January–August 2020 [16] showed that the PYLL/death from confirmed COVID-19 cases ranged from 19.3 years (Ukraine) to 2.7 years (Australia), with countries with higher values (e.g., Peru—15.2 years, Colombia 14.9 years, Cape Verde 14.7 years, etc.) compared to the lower values recorded in other countries (e.g., Slovenia—2,8 years, Sweden 3.2 years, Norway 4.0 years, France 4.8 years, Georgia 6.2 years); these results refer to different periods since the first case occurred in each country, and do not cover the entire year of 2020 but use 80 years as the age limit for the PYLL calculation. 

In 2021, the total number of deaths rose to 39,966, with 28,777 premature deaths, leading to a higher PYLL of 386,061. The PYLL per death ratio remained relatively constant at 13.42 (Table 2). The variation in PYLL/death in the counties of Romania in 2021, calculated according to our study, is between 11.83 years (Cluj County) and 15.03 (Ilfov County).

In confirmed COVID-19 cases, the proportion of premature deaths was relatively constant in the analyzed years (72.74% in 2020 and 72.00% in 2021); there was an increase in the number of premature deaths by 103.3% in 2021 compared to 2020, and the number of PYLL increased by 99.52% in 2021 compared to 2020 (Table 1 and Table 2).

The proportion of premature deaths (<80 years) to confirmed COVID-19 cases in Romania was 72.74% (in 2020) and 72.74% (in 2021), with the counties varying between 61.27% (Cluj County) and 84.45% (Olt County) in 2020, respectively, and between 65.71% (Cluj County) and 79.62% (Maramureș County) in 2021 (Figure 2). The percentage differences observed between 2020 and 2021 are statistically significant in three counties: Hunedoara, Olt and Prahova (*p* < 0.05). For the rest of the counties, the difference is not statistically significant. The male gender is the most affected by premature mortality at the national and county level, except for some counties in 2021 where the proportion of female deaths was higher in women (Satu Mare, Salaj, Calarasi, Alba), according to Figure 3 and Figure 4. 

In 2020, the crude rate of PYLL for both genders recorded the highest value in Bihor County (1698.49/100,000 inhabitants) and the lowest in Giurgiu County (308.5/100,000 inhabitants), the other counties being between the two extreme values (Table 1). In the same year, the crude PYLL rate by gender ranged between 1325.16/100,000 inhabitants (Bihor County) and 228.90/100,000 inhabitants (Giurgiu County), and ranged between 2215.13/100,000 inhabitants (Sibiu County) and 385.15/100,000 inhabitants (Giurgiu County) for males, as shown in Table 3. There is a statistically significant difference between men and women at the national level in 2020 regarding the analyzed indicators, with a predominant impact on males.

In 2021, the crude rate of PYLL, for both genders, recorded the highest value in Caraș-Severin County (3553.65/100,000 inhabitants) and the lowest in Ilfov County (1365.12/100,000 inhabitants), the other counties being between the two extreme values (Table 2). In the same year, the crude PYLL rate by gender was between 3761.77/100,000 inhabitants (Caraș-Severin county) and 1606.38/100,000 inhabitants (Ilfov county), and between 3350.2/100,000 inhabitants (Caraș-Severin county) and 1130.94/100,000 inhabitants (Ilfov county), as shown in Table 4. There is a significant difference between men and women at the national level in 2021 regarding the analyzed indicators, with a greater impact on males.

The standardized rates of PYLL for both genders, which eliminate differences in the population distribution by age group, and their representation in map form provide a visual representation of the geographical distribution of the impact of COVID-19 deaths (Figure 5 and Figure 6). For each year analyzed, counties can be ranked by the impact caused by years of life lost to confirmed COVID-19 cases and the increase in the standardized rate of PYLL recorded in 2020 and 2021 in all counties. These increases may show an intensification of the impact of the pandemic in the second year, a higher detection rate and/or a lower effectiveness of control and prevention measures. The counties that saw the highest increase in the standardized PYLL rate in 2021 compared to 2020 were Giurgiu (5.89-fold increase), Satu Mare (4.37-fold increase), Dolj (3.56-fold increase) and Covasna (3.38-fold), and the counties with the lowest increases were Mures (1.53-fold increase), Suceava (1.42-fold increase) and Arges (1.21-fold increase). The age-standardized rates of PYLL (per 100,000 inhabitants) for confirmed cases of COVID-19 varied, in 2020 for Romanian counties, between 301 (Giurgiu County) and 1696 (Bihor County), and in 2021, between 1382 (Buzau County) and 3251 (Bihor County). 

According to the study carried out in 17 countries up to week 35 in 2020 [16], the age-standardized rates of PYLL (per 100,000 inhabitants) in confirmed cases of COVID-19 ranged from 1377.0 (Peru) to 2.0 (Georgia); some countries had high values (e.g., Brazil—678.0, Colombia—567.0, USA—343.0) and others had low values (e.g., Norway—24.0, Slovenia—9.0, Cyprus—17.0).

Across all counties of Romania, the burden of premature mortality in confirmed cases of COVID-19 in 2021 was higher compared to 2020. This increase, as reflected by the rise in the total PYLL and the standardised rate of PYLL, occurred in the face of two pandemic waves in each analyzed year. The magnitude of this increase, varying from one county to another, underlines the complex interaction between the population’s vulnerability (which was higher after the first year of the pandemic) and the factors that came into play in the second year, such as the continuation of social restrictions (despite the possibility of limited social activity), limited access to curative and preventive health services, and the emergence of SARS-CoV-2 virus mutations.

### Study Limitations

The impossibility of establishing the county of infection for the 29 deaths out of the 65,406 analyzed (Figure 2) is one of the limitations of this study, but it has a minimal impact considering the 59,421 cases analyzed to calculate PYLL. Another possible limitation to the registration of deaths due to COVID-19 was determined by the variable testing capacity from one county to another: counties that developed testing laboratories as early as possible at the beginning of the pandemic benefited from the optimal diagnosis of infection cases and consequently the correct registration of the cause of death. 

This study analyzes deaths and PYLL by calendar year without considering the four waves of the COVID-19 pandemic (the first two waves in 2020 and the third and fourth waves in 2021) that affected Romania [42,43]. Also, age matching was not used in the statistical analysis.

This study’s threshold for calculating PYLL (80 years) was chosen because some authors [16,30] consider it more reliable in the case of deaths caused by COVID-19 (even though life expectancy at birth in Romania is lower than this limit) and because it allows comparison with international data. This study did not consider the impact of comorbidities and vaccination on deaths in people infected with COVID-19, nor the economic impact of premature deaths, expressed in PYLL, in the population studied.

## 5. Conclusions

In 2021, Romania recorded a marked increase in the total number of deaths and premature deaths caused by COVID-19. With a 2.03-fold increase in the number of premature deaths compared to the previous year, the country also saw a 1.99-fold increase in Potential Years of Life Lost (PYLL). Despite the increase in deaths and PYLL in 2021, the PYLL per death ratio remained relatively constant at (13.42 in 2021 vs. 13.67 in 2020). This suggests consistency in the average lifespan lost for each COVID-19 death.

The analyzed data indicate that the male gender is more affected by premature mortality nationally and in most counties, except for specific situations in 2021.

The analysis of crude and standardized rates of PYLL across both genders reveals significant variations between counties. Most regions saw significant increases in the standardized rate of PYLL in 2021 compared to 2020 in confirmed COVID-19 cases. These variations can be influenced by the different pandemic management strategies, available resources, and demographic peculiarities of each county.

Overall, the data provided and analyzed in this article provide a complex picture of the impact of the COVID-19 pandemic in Romania, highlighting the need for effective management strategies adapted to the specifics of each county to reduce the fatality of cases and consequently the potential years of life lost.

## Figures and Tables

**Figure 1 healthcare-12-01189-f001:**
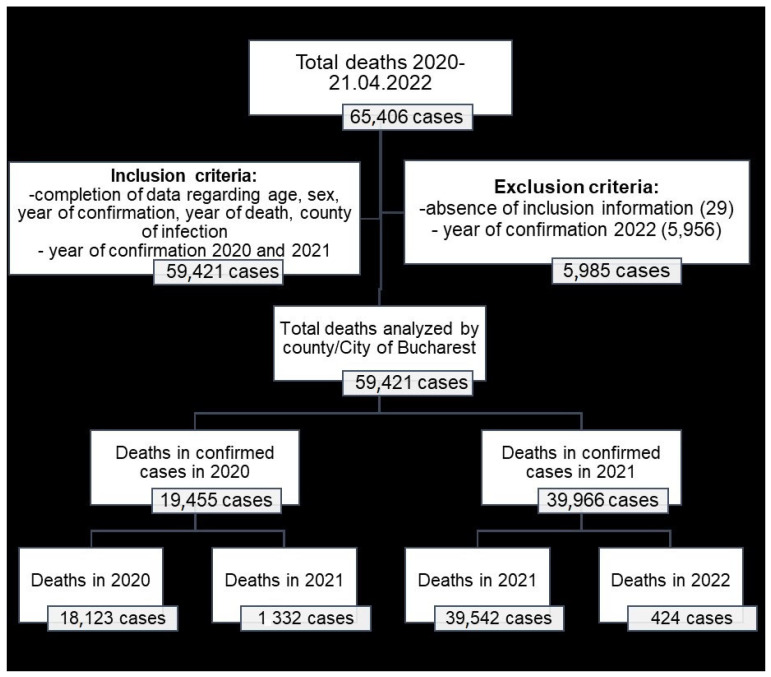
Study population selection.

**Figure 2 healthcare-12-01189-f002:**
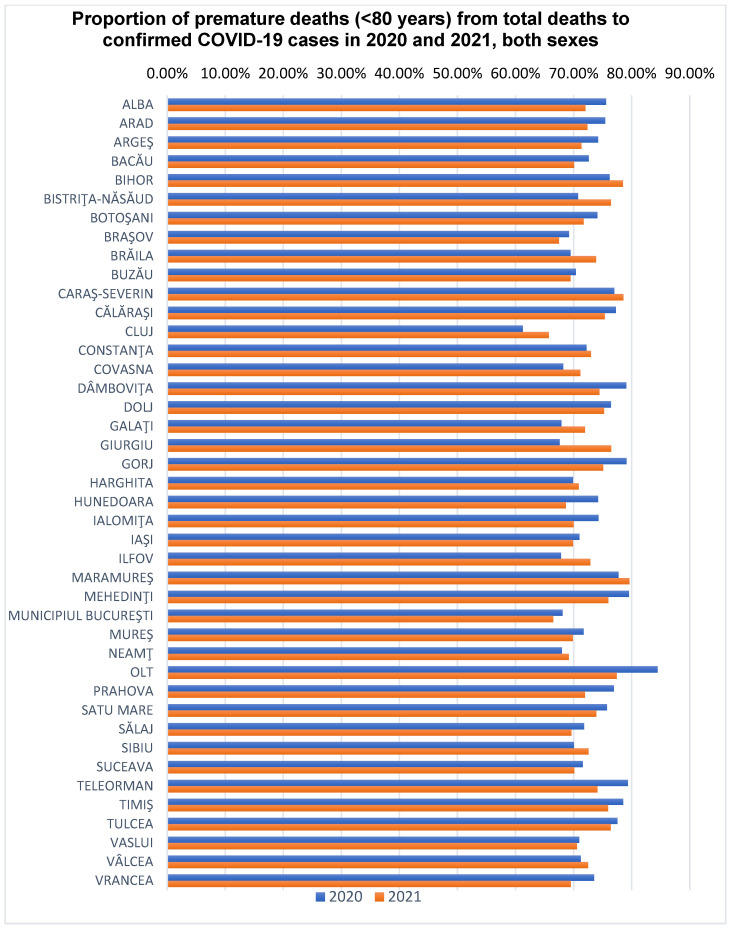
The proportion of premature deaths (<80 years) of the total deaths from confirmed COVID-19 cases, for both genders, by Romanian counties. Notes: Between the two years calculated using the chi-squared test; *p* < 0.05 was obtained in the case of Hunedoara counties (0.04463) and Olt (0.0214); *p* < 0.01 for Prahova County (0.006594); for the other counties *p* > 0.05.

**Figure 3 healthcare-12-01189-f003:**
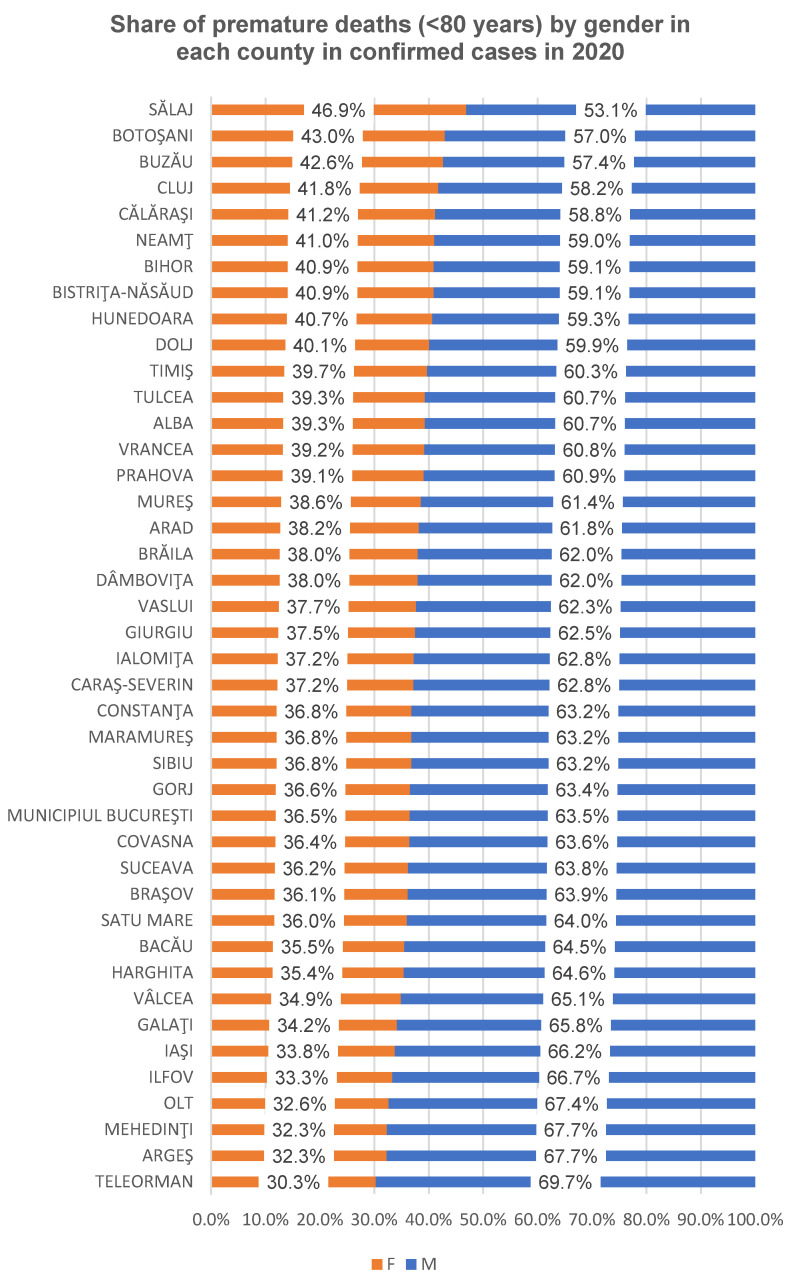
Sex ratio of premature deaths (<80 years) in each county to confirmed COVID-19 cases in 2020.

**Figure 4 healthcare-12-01189-f004:**
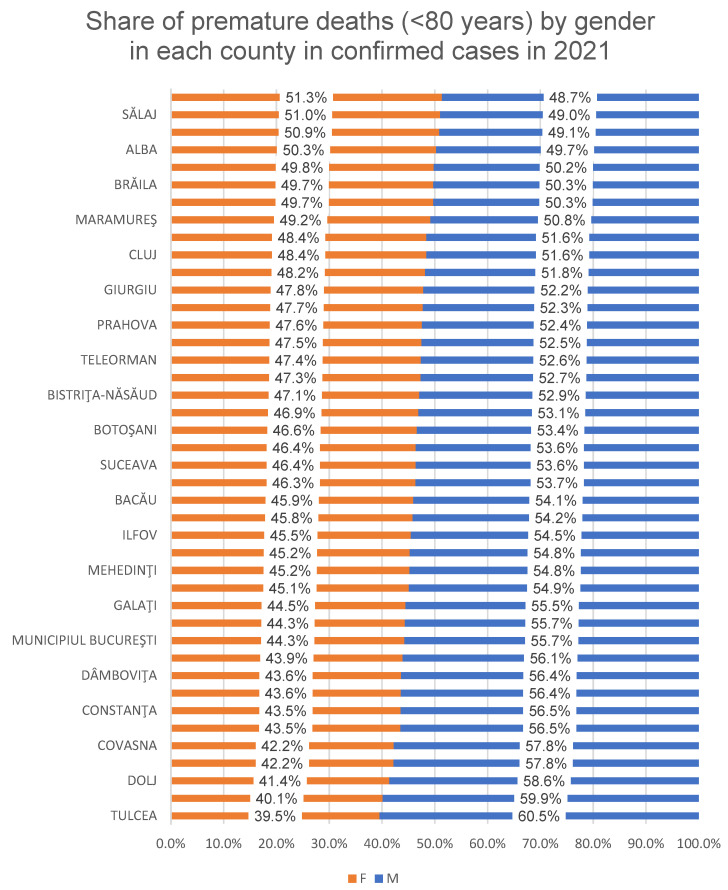
Sex ratio of premature deaths (<80 years) in each county to confirmed COVID-19 cases in 2021.

**Figure 5 healthcare-12-01189-f005:**
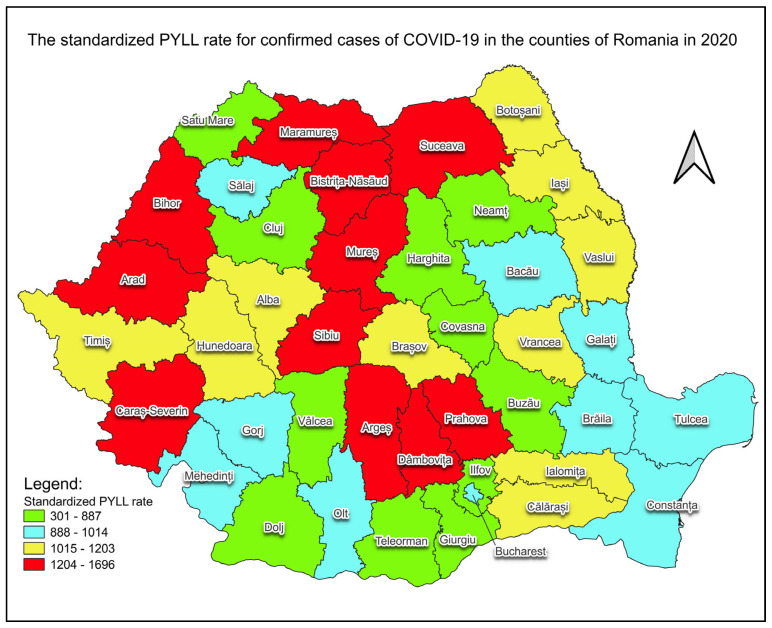
PYLL’s standardized rate of confirmed COVID–19 cases, for both genders, in 2020. Map created using the Free and Open Source QGIS.

**Figure 6 healthcare-12-01189-f006:**
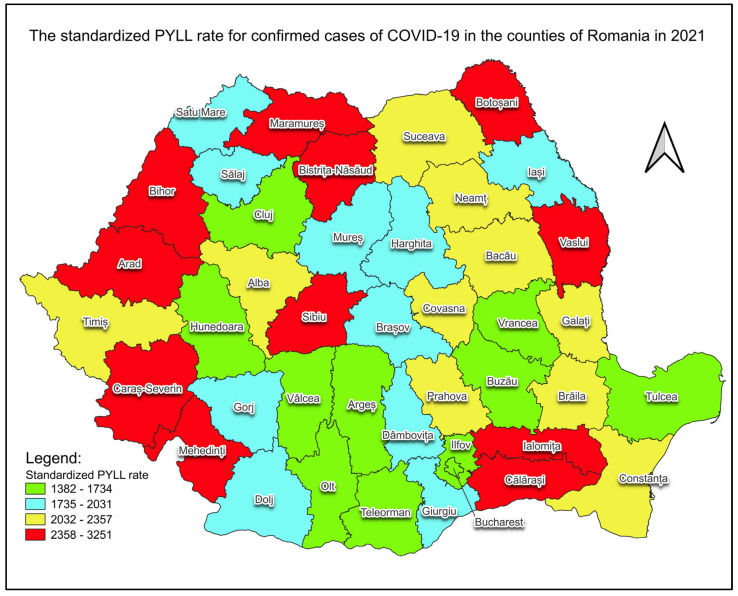
PYLL’s standardized rate of confirmed COVID-19 cases, for both genders, in 2021. Map created using the Free and Open Source QGIS.

**Figure 7 healthcare-12-01189-f007:**
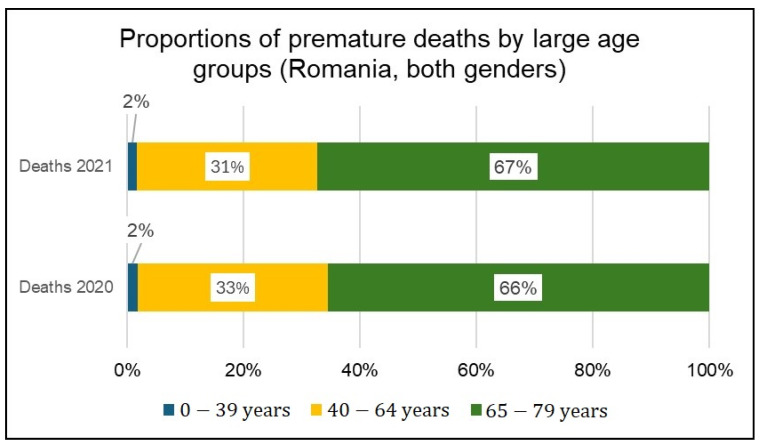
Proportion of premature deaths by age groups (both genders) in 2020 and 2021.

**Figure 8 healthcare-12-01189-f008:**
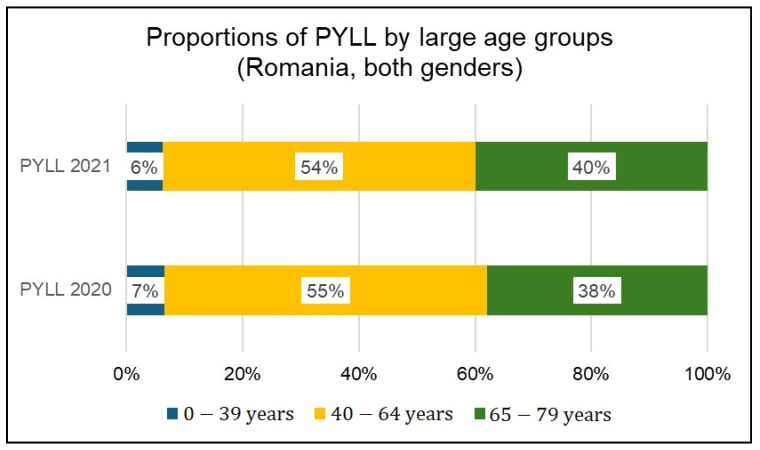
Proportion of PYLL by age groups (both genders) in 2020 and 2021.

**Table 1 healthcare-12-01189-t001:** Number of deaths and PYLL according to the counties of Romania and Bucharest in confirmed COVID-19 cases in both genders in 2020.

Administrative-Territorial Unit	Deaths—All Ages		Premature Deaths		PYLL		PYLL/ Death
	Number	Crude Rate *	Number	Crude Rate *	Number	Crude Rate *	
ALBA	340	105.35	257	84.05	3418	1117.85	13.30
ARAD	607	146.42	458	115.64	6395	1614.68	13.96
ARGEŞ	768	134.35	570	104.93	7966	1466.50	13.98
BACĂU	562	96.95	408	74.07	5502	998.83	13.49
BIHOR	811	144.99	618	115.06	9123	1698.49	14.76
BISTRIŢA-NĂSĂUD	366	131.95	259	97.34	3286	1234.94	12.69
BOTOŞANI	355	94.53	263	74.32	3701	1045.86	14.07
BRAŞOV	688	124.19	476	89.79	6489	1224.07	13.63
BRĂILA	311	109.70	216	81.20	2779	1044.71	12.87
BUZĂU	260	63.92	183	48.04	2835	744.18	15.49
CARAŞ-SEVERIN	370	137.91	285	111.56	3770	1475.78	13.23
CĂLĂRAŞI	264	94.49	204	77.44	2929	1111.93	14.36
CLUJ	645	90.80	395	58.23	4514	665.42	11.43
CONSTANŢA	688	102.61	497	77.14	6332	982.82	12.74
COVASNA	173	86.12	118	61.24	1213	629.51	10.28
DÂMBOVIŢA	483	99.49	382	82.70	5631	1219.11	14.74
DOLJ	297	47.94	227	38.57	3141	533.63	13.84
GALAŢI	539	107.47	366	76.58	4580	958.29	12.51
GIURGIU	71	27.07	48	19.46	761	308.50	15.85
GORJ	273	88.04	216	73.12	2915	986.77	13.50
HARGHITA	279	92.87	195	67.84	2364	822.37	12.12
HUNEDOARA	454	120.26	337	93.99	4253	1186.17	12.62
IALOMIŢA	253	99.61	188	78.19	2712	1127.86	14.43
IAŞI	721	90.56	512	67.05	6797	890.15	13.28
ILFOV	283	57.13	192	40.08	3088	644.66	16.08
MARAMUREŞ	566	123.81	440	100.04	5834	1326.40	13.26
MEHEDINŢI	210	88.35	167	74.14	2460	1092.12	14.73
BUCHAREST	1774	96.68	1208	68.75	16,189	921.30	13.40
MUREŞ	658	123.78	472	93.11	6685	1318.75	14.16
NEAMŢ	384	87.95	261	63.60	3725	907.72	14.27
OLT	283	72.93	239	65.35	3673	1004.35	15.37
PRAHOVA	898	126.79	691	103.35	9354	1399.05	13.54
SATU MARE	132	39.93	100	31.26	1317	411.69	13.17
SĂLAJ	181	86.34	130	65.20	1843	924.31	14.18
SIBIU	628	156.64	440	114.24	6227	1616.77	14.15
SUCEAVA	869	139.23	622	104.87	8539	1439.75	13.73
TELEORMAN	179	54.77	142	47.11	2174	721.18	15.31
TIMIŞ	648	91.70	509	74.72	7004	1028.18	13.76
TULCEA	187	97.87	145	79.38	1908	1044.55	13.16
VASLUI	389	104.76	276	78.34	3962	1124.60	14.36
VÂLCEA	306	88.09	218	68.20	2894	905.31	13.28
VRANCEA	302	95.50	222	74.60	3207	1077.66	14.45
Total—Romania	19,455	100.82	14,152	77.04	193,489	1053.28	13.67

* Per 100,000 inhabitants.

**Table 2 healthcare-12-01189-t002:** Number of deaths and PYLL according to the counties of Romania and Bucharest in confirmed COVID-19 cases in both genders in 2021.

Administrative-Territorial Unit	Deaths—All Ages		Premature Deaths		PYLL		PYLL/ Death
	Number	Crude Rate *	Number	Crude Rate *	Number	Crude Rate *	
ALBA	815	252.00	587	191.00	7749	2521.40	13.20
ARAD	1104	268.13	799	202.70	10,590	2686.58	13.25
ARGEŞ	1065	187.70	760	140.68	9697	1794.93	12.76
BACĂU	1253	213.24	878	156.82	11,906	2126.58	13.56
BIHOR	1581	284.99	1241	232.63	17,262	3235.83	13.91
BISTRIŢA-NĂSĂUD	670	234.35	512	186.18	6707	2438.90	13.10
BOTOŞANI	856	224.34	614	170.16	8318	2305.20	13.55
BRAŞOV	1196	217.12	807	152.98	10,488	1988.23	13.00
BRĂILA	716	255.23	529	200.56	6906	2618.22	13.05
BUZĂU	619	153.27	430	113.35	5833	1537.65	13.57
CARAŞ-SEVERIN	794	309.67	624	255.47	8680	3553.65	13.91
CĂLĂRAŞI	621	221.83	468	176.60	6659	2512.76	14.23
CLUJ	1321	189.59	868	130.49	10,271	1544.09	11.83
CONSTANŢA	1580	238.63	1153	181.19	15,063	2367.06	13.06
COVASNA	416	207.91	296	154.13	4181	2177.11	14.13
DÂMBOVIŢA	881	183.18	656	143.02	9137	1992.06	13.93
DOLJ	1066	175.37	802	138.72	11,058	1912.72	13.79
GALAŢI	1141	230.42	821	173.81	11,506	2435.92	14.01
GIURGIU	421	161.47	322	130.77	4556	1850.33	14.15
GORJ	594	191.25	446	150.39	6515	2196.86	14.61
HARGHITA	601	203.42	426	150.54	5412	1912.46	12.70
HUNEDOARA	795	216.43	546	156.47	6515	1867.08	11.93
IALOMIŢA	667	265.45	467	195.92	6891	2891.04	14.76
IAŞI	1375	176.21	961	128.33	12,717	1698.21	13.23
ILFOV	634	120.86	462	90.80	6946	1365.12	15.03
MARAMUREŞ	1050	231.65	836	191.57	11,640	2667.34	13.92
MEHEDINŢI	658	279.83	500	223.79	7015	3139.81	14.03
BUCHAREST	3378	190.50	2246	132.35	29,250	1723.65	13.02
MUREŞ	1045	199.33	730	145.90	10,114	2021.43	13.85
NEAMŢ	1070	241.75	740	177.40	9991	2395.12	13.50
OLT	554	144.32	429	118.17	6106	1681.92	14.23
PRAHOVA	1802	257.78	1297	196.20	16,840	2547.43	12.98
SATU MARE	606	183.68	448	140.19	5941	1859.13	13.26
SĂLAJ	507	241.10	353	176.05	4231	2110.06	11.99
SIBIU	1036	262.51	752	198.26	9751	2570.77	12.97
SUCEAVA	1295	205.38	908	151.13	12,228	2035.31	13.47
TELEORMAN	587	181.56	435	145.08	5759	1920.74	13.24
TIMIŞ	1243	183.07	944	144.30	13,236	2023.26	14.02
TULCEA	318	166.29	243	132.56	3138	1711.81	12.91
VASLUI	877	236.62	619	175.61	8315	2358.94	13.43
VÂLCEA	571	166.05	414	129.32	5385	1682.05	13.01
VRANCEA	587	181.00	408	133.08	5558	1812.83	13.62
Total—Romania	39,966	208.96	28,777	157.77	386,061	2116.63	13.42

* Per 100,000 inhabitants.

**Table 3 healthcare-12-01189-t003:** PYLL indicators by gender in 2020.

Administrative-Territorial Unit	Premature Deaths—Number		PYLL		Crude Rate of PYLL per 100,000 Inhabitants		PYLL/Death		*t*-Test
	Female	Male	Female	Male	Female	Male	Female	Male	*p* Value
ALBA	101	156	1227	2191	804.38	1429.92	12.15	14.04	0.1
ARAD	175	283	2600	3795	1.300.01	1935.69	14.86	13.41	0.1
ARGEŞ	184	386	2579	5387	942.54	1998.31	14.02	13.96	0.9
BACĂU	145	263	1838	3664	669.32	1326.39	12.68	13.93	0.2
BIHOR	253	365	3591	5532	1.325.16	2078.61	14.19	15.16	0.3
BISTRIŢA-NĂSĂUD	106	153	1292	1994	977.56	1488.95	12.19	13.03	0.5
BOTOŞANI	113	150	1485	2216	853.02	1232.60	13.14	14.77	0.2
BRAŞOV	172	304	2189	4300	814.89	1644.40	12.73	14.14	0.1
BRĂILA	82	134	1017	1762	766.54	1321.50	12.40	13.15	0.5
BUZĂU	78	105	1147	1688	605.10	881.94	14.71	16.08	0.4
CARAŞ-SEVERIN	106	179	1328	2442	1.032.28	1925.70	12.53	13.64	0.4
CĂLĂRAŞI	84	120	1062	1867	810.87	1409.65	12.64	15.56	0.03
CLUJ	165	230	1797	2717	517.54	820.49	10.89	11.81	0.3
CONSTANŢA	183	314	2069	4263	630.85	1347.78	11.31	13.58	0.007
COVASNA	43	75	393	820	409.17	848.49	9.14	10.93	0.3
DÂMBOVIŢA	145	237	1972	3659	862.08	1569.40	13.60	15.44	0.08
DOLJ	91	136	1133	2008	382.61	686.53	12.45	14.76	0.1
GALAŢI	125	241	1400	3180	584.58	1333.63	11.20	13.20	0.06
GIURGIU	18	30	277	484	228.90	385.15	15.39	16.13	0.8
GORJ	79	137	1059	1856	719.76	1251.72	13.41	13.55	0.9
HARGHITA	69	126	904	1460	629.91	1014.25	13.10	11.59	0.3
HUNEDOARA	137	200	1666	2587	924.37	1450.78	12.16	12.94	0.4
IALOMIŢA	70	118	897	1815	751.00	1499.83	12.81	15.38	0.09
IAŞI	173	339	2003	4794	528.77	1245.92	11.58	14.14	0.006
ILFOV	64	128	765	2323	317.04	977.22	11.95	18.15	<0.00001
MARAMUREŞ	162	278	2149	3685	970.44	1687.33	13.27	13.26	1
MEHEDINŢI	54	113	700	1760	623.55	1557.66	12.96	15.58	0.1
BUCHAREST	441	767	5513	10,676	594.19	1287.23	12.50	13.92	0.01
MUREŞ	182	290	2590	4095	1019.11	1620.02	14.23	14.12	0.9
NEAMŢ	107	154	1591	2134	776.87	1038.09	14.87	13.86	0.4
OLT	78	161	1243	2430	684.25	1320.29	15.94	15.09	0.5
PRAHOVA	270	421	3627	5727	1078.51	1723.44	13.43	13.60	0.8
SATU MARE	36	64	494	823	304.10	522.70	13.72	12.86	0.7
SĂLAJ	61	69	797	1046	798.31	1050.65	13.07	15.16	0.3
SIBIU	162	278	2022	4205	1035.23	2215.13	12.48	15.13	0.01
SUCEAVA	225	397	2985	5554	1014.27	1858.84	13.27	13.99	0.4
TELEORMAN	43	99	518	1656	348.77	1082.88	12.05	16.73	0.01
TIMIŞ	202	307	2691	4313	777.34	1287.37	13.32	14.05	0.4
TULCEA	57	88	678	1230	751.19	1331.08	11.89	13.98	0.1
VASLUI	104	172	1483	2479	862.53	1374.42	14.26	14.41	0.9
VÂLCEA	76	142	965	1929	605.06	1204.28	12.70	13.58	0.5
VRANCEA	87	135	1220	1987	817.84	1338.80	14.02	14.72	0.6
Total—Romania	5308	8844	68,956	124,533	745.36	1365.69	12.99	14.08	<0.00001

**Table 4 healthcare-12-01189-t004:** PYLL indicators by gender in 2021.

Administrative-Territorial Unit	Premature Deaths—Number		PYLL		Crude Rate of PYLL per 100,000 Inhabitants		PYLL/Death		*t*-Test
	Female	Male	Female	Male	Female	Male	Female	Male	*p* Value
ALBA	295	292	4001	3748	2603.14	2439.63	13.56	12.84	0.3
ARAD	397	402	4955	5635	2483.16	2895.13	12.48	14.02	0.02
ARGEŞ	331	429	4049	5648	1480.90	2116.70	12.23	13.17	0.2
BACĂU	403	475	5265	6641	1878.89	2374.76	13.06	13.98	0.2
BIHOR	618	623	8424	8838	3119.13	3355.48	13.63	14.19	0.3
BISTRIŢA-NĂSĂUD	241	271	3166	3541	2312.21	2564.53	13.14	13.07	0.9
BOTOŞANI	286	328	3753	4565	2110.50	2494.37	13.12	13.92	0.3
BRAŞOV	374	433	4612	5876	1717.62	2268.78	12.33	13.57	0.055
BRĂILA	263	266	3412	3494	2572.88	2664.06	12.97	13.14	0.8
BUZĂU	187	243	2316	3517	1219.14	1857.15	12.39	14.47	0.04
CARAŞ-SEVERIN	302	322	4138	4542	3350.20	3761.77	13.70	14.11	0.6
CĂLĂRAŞI	238	230	3234	3425	2445.76	2579.49	13.59	14.89	0.2
CLUJ	420	448	4578	5693	1344.42	1753.51	10.90	12.71	0.004
CONSTANŢA	502	651	6294	8769	1937.71	2814.72	12.54	13.47	0.1
COVASNA	125	171	1651	2530	1716.48	2639.29	13.21	14.80	0.2
DÂMBOVIŢA	286	370	3710	5427	1622.35	2359.66	12.97	14.67	0.02
DOLJ	332	470	4312	6746	1473.54	2362.86	12.99	14.35	0.055
GALAŢI	365	456	4941	6565	2077.92	2798.85	13.54	14.40	0.2
GIURGIU	154	168	2208	2348	1813.95	1885.88	14.34	13.98	0.8
GORJ	179	267	2608	3907	1760.97	2631.70	14.57	14.63	0.9
HARGHITA	187	239	2118	3294	1494.78	2331.33	11.33	13.78	0.01
HUNEDOARA	263	283	3068	3447	1733.91	2004.07	11.67	12.18	0.5
IALOMIŢA	219	248	2937	3954	2472.39	3306.99	13.41	15.94	0.008
IAŞI	440	521	5520	7197	1477.14	1918.43	12.55	13.81	0.04
ILFOV	210	252	2920	4026	1130.94	1606.38	13.90	15.98	0.04
MARAMUREŞ	411	425	5434	6206	2466.64	2871.96	13.22	14.60	0.054
MEHEDINŢI	226	274	3124	3891	2791.28	3489.65	13.82	14.20	0.7
BUCHAREST	994	1252	12,112	17,138	1345.56	2150.74	12.19	13.69	<0.00001
MUREŞ	329	401	4247	5867	1687.08	2359.98	12.91	14.63	0.04
NEAMŢ	328	412	4141	5850	1975.26	2819.32	12.63	14.20	0.02
OLT	181	248	2533	3573	1402.00	1959.25	13.99	14.41	0.7
PRAHOVA	617	680	7455	9385	2228.32	2874.41	12.08	13.80	0.001
SATU MARE	230	218	2793	3148	1718.35	2004.87	12.14	14.44	0.01
SĂLAJ	180	173	2033	2198	2017.61	2203.44	11.29	12.71	0.1
SIBIU	359	393	4589	5162	2380.21	2767.77	12.78	13.13	0.6
SUCEAVA	421	487	5380	6848	1795.85	2273.47	12.78	14.06	0.054
TELEORMAN	206	229	2467	3292	1658.50	2178.92	11.98	14.38	0.006
TIMIŞ	448	496	5941	7295	1780.51	2275.96	13.26	14.71	0.02
TULCEA	96	147	1333	1805	1461.54	1959.61	13.89	12.28	0.2
VASLUI	280	339	3665	4650	2125.03	2583.03	13.09	13.72	0.4
VÂLCEA	192	222	2511	2874	1741.22	1508.33	13.08	12.95	0.9
VRANCEA	193	215	2534	3024	1927.73	1580.05	13.13	14.07	0.3
Total—Romania	13,308	15,469	170,482	215,579	2320.00	1787.99	12.81	13.94	<0.00001

## Data Availability

Data are contained within the article.

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
