# Peer review of "Using Potential Years of Life Lost (PYLL) to Compare Premature Mortality between Romanian Counties to Confirmed COVID-19 Cases in 2020 and 2021"

_healthcare, 2024, doi:10.3390/healthcare12121189_

Round 1

Reviewer 1 Report

Comments and Suggestions for Authors

This is a good manuscript considering the scientific content and impact on human health. However, I found that the in discussion section narrative of results of the study. There was no comparison of the study findings with other study done other countries. There was also no discussion why the study findings were different or not within counties. Significance of the study findings were not discussed in regard to economic loss with PYLL or YLL. My opinion result and discussion need not to be separate rather it can be "Result and discussion". 

Author Response

Thank you for your review. We tried, based on your recommendations, to improve the content of the article.

Reviewer 2 Report

Comments and Suggestions for Authors

Thank you for inviting me to review the manuscript entitled "Using potential years of life lost (PYLL) to compare premature mortality between Romanian counties to confirmed COVID-19 cases in 2020 and 2021." The manuscript analyses COVID-19-related deaths that occurred in 2020 and 2021 and their PYLL. 

While the comparative study is intriguing, it is my opinion that further elaboration and clarification are required.

First, it is critical to comprehend why the authors rely on the European age standard of 80 years rather than the considerable gender differences in Romanian standards (75.4 years on average for men and 79.1 years for women). This roughly 5-year disparity, particularly in male life expectancy, raises my concerns. It is probable that analyses conducted according to Romanian standards would produce different results; therefore, I ask the Authors to provide a justification for employing the 80-year threshold, which may be regarded as a limitation of the study and a possible source of bias.

Another element is the lack of an assessment regarding PYLL in relation to COVID-19 vaccinations. Undoubtedly, assessing this aspect would yield a more comprehensive overview of the differences in the rate of PYLL growth or decline across counties. It is anticipated that counties experiencing a decline in this trend will have a higher proportion of vaccinated individuals, or premature fatalities attributable to unvaccinated individuals against COVID-19 will be more prevalent. Given the potential accessibility of a ministerial database and the assumption that such a database includes COVID-19 vaccinations, I encourage the authors to devise a method that complies with privacy regulations in order to obtain information regarding vaccination status (i.e., no vaccination, one dose, or two doses) and integrate this variable into statistical analyses. Alternatively, should it be unfeasible to obtain access to this information, aggregated data concerning the proportions of individuals who received the COVID-19 vaccine in 2021 are presumably available for each county. In this case, however, the results represent the entire population and not just the subjects under investigation; thus, they should be interpreted with caution.

Additional minor details comprise the omission of the initial "C" from the misspelt "chi-square" and the use of Romanian terms within the tables (e.g., "municipul bucuresti").

Author Response

(The authors gave the same response as above.)

Reviewer 3 Report

Comments and Suggestions for Authors

Summary: The authors tried to investigate the variation in Potential Life Lost Years (PYLL) caused by COVID-19 in Romanian counties in 2020 and 2021. Their results highlighted the need for effective management strategies to reduce the fatality of cases in specific counties for potential future epidemics.

Comments: 1. The authors did not comment on the age of male and female cases in different counties and if they use age-matching in their statistical analysis. 

2. Authors did not comment on the different phases on COVID-19 and its impact on their analysis of the rate of PYLL.

Author Response

(The authors gave the same response as above.)
